# Proximal Tibiofibular Dislocation in Closing-Wedge High Tibial Osteotomy Increases the Risk of Medium and Long-Term Total Knee Replacement

**DOI:** 10.3390/jcm10132743

**Published:** 2021-06-22

**Authors:** Juan Sánchez-Soler, Alex Coelho, Raúl Torres-Claramunt, Berta Gasol, Albert Fontanellas, Simone Perelli, Pedro Hinarejos, Joan Carles Monllau

**Affiliations:** 1Department of Surgery and Morphologic Science, Orthopaedic Surgery Service, Universitat Autònoma de Barcelona, Hospital del Mar, 08003 Barcelona, Spain; 63611@parcdesalutmar.cat (A.C.); rtorresclaramunt@psmar.cat (R.T.-C.); bgasolcudos@gmail.com (B.G.); 64292@parcdesalutmar.cat (A.F.); sperelli@parcdesalutmar.cat (S.P.); phinarejos@hospitaldelmar.cat (P.H.); jmonllau@parcdesalutmar.cat (J.C.M.); 2IMIM (Hospital del Mar Medical Research Institute), 08003 Barcelona, Spain; 3Catalan Institute for Traumatology and Sports Medicine (ICATME), Hospital Universitari Dexeus, UAB, 08028 Barcelona, Spain

**Keywords:** closing-wedge osteotomy, knee stability, tibiofibular dislocation, fibular osteotomy, high tibial osteotomy, osteotomy survival rate, total knee replacement, knee

## Abstract

Proximal tibiofibular dislocation in closing-wedge high tibial osteotomy increases the risk of medium and long-term total knee replacement. *Background*: High tibial osteotomy is an effective treatment for medial osteoarthritis in young patients with varus knee. The lateral closing-wedge high tibial osteotomy (CWHTO) may be managed with tibiofibular dislocation (TFJD) or a fibular head osteotomy (FHO). TFJD may lead to lateral knee instability and thereby affect mid- and long-term outcomes. It also brings the osteotomy survival rate down. *Objective*: To compare the CWHTO survival rate in function of tibiofibular joint management with TFJD or FHO, and to determine whether medium and long-term clinical outcomes are different between the two procedures. *Material & Methods*: A retrospective cohort study was carried out that included CWHTO performed between January 2005 to December 2018. Those patients were placed in either group 1 (FHO) or Group 2 (TFJD). Full-leg weight-bearing radiographs were studied preoperatively, one year after surgery and at final follow-up to assess the femorotibial angle (FTA). The Rosenberg view was used to assess the Ahlbäck grade. The Knee Society Score (KSS) was used to assess clinical outcomes and a Likert scale for patient satisfaction. The total knee replacement (TKR) was considered the end of the follow-up and the point was to analyze the CWHTO survival rate. A sub-analysis of both cohorts was performed in patients who had not been FTA overcorrected after surgery (postoperative FTA ≤ 180°, continuous loading in varus). *Results*: A total of 230 knees were analyzed. The follow-up period ranged from 24–180 months. Group 1 (FHO) consisted of 105 knees and group 2 (TFJD) had 125. No preoperative differences were observed in terms of age, gender, the KSS, FTA or the Ahlbäck scale; neither were there any differences relative to postop complications. The final follow-up FTA was 178.7° (SD 4.9) in group 1 and 179.5° (SD 4.2) in group 2 (*p* = 0.11). The Ahlbäck was 2.21 (SD 0.5) in group 1 and 2.55 (SD 0.5) in group 2 (*p* = 0.02) at the final follow-up. The final KSS knee values were similar for group 1 (86.5 ± 15.9) and group 2 (84.3 ± 15.8). Although a non-significant trend of decreased HTO survival in the TFJD group was found (*p* = 0.06) in the sub-analysis of non-overcorrected knees, which consisted of 52 patients from group 1 (FHO) and 58 from group 2 (TFJD), 12.8% of the patients required TKR with a mean of 88.8 months in group 1 compared to 26.8% with a mean of 54.9 months in the case of group 2 (*p* = 0.005). However, there were no differences in clinical and radiological outcomes. *Conclusion*: TFJD associated with CWHTO shows an increase in the conversion to TKR at medium and long-term follow-up with lower osteotomy survival than the CWHTO associated with FHO, especially in patients with a postoperative FTA ≤ 180° (non-overcorrected). There were no differences in clinical, radiological or satisfaction results in patients who did not require TKR. Level of evidence III. Retrospective cohort study.

## 1. Introduction

High tibial osteotomy (HTO) is a joint-preserving procedure that is widely accepted as an effective treatment for young patients with isolated medial compartment osteoarthritis (OA) in varus knee [1,2]. The purpose of the procedure is to transfer weight-bearing forces from the medial to the lateral knee compartment to reduce the load and contact area over the previously affected compartment. The most commonly used techniques include the lateral closing-wedge HTO (CWHTO) and the medial open-wedge HTO (OWHTO) [3]. Over recent years, OWHTO has gained popularity for the treatment of symptomatic varus knees. However, there is some controversy as to whether there are functional and radiological differences between the two procedures [4,5].

Although OWHTO has been associated with higher non-union rates and donor site morbidity, one of the main reported disadvantages of CWHTO is tibiofibular joint (TFJ) manipulation with either TFJ dislocation (TFJD) or fibular osteotomy [6,7]. One of the major perioperative complications after fibular head osteotomy (FHO) is peroneal nerve disfunction. The reported incidence of symptomatic injury stands at between 3–20% [8]. On the other hand, since lateral collateral ligament (LCL) and popliteus-fibular ligament (PFL) originate on the fibular head, TFJD may lead to fibular head rise and a shift in ligament tensioning with a potential impact on lateral knee laxity [9].

Scarce data has been published about lateral knee stability that compares the two procedures used for FTJ manipulation. Torres-Claramunt et al. reported that lateral knee compartment gapping is greater when a TFJD is performed instead of a fibular head osteotomy (FHO) at 1 year after an HTO without affecting clinical results [9]. 

Different factors have been described that lower the CWHTO survival rate. They include preoperative osteoarthritis grade >2 (Ahlbäck), female gender, obesity and being over 50 years old. However, there is no data analyzing whether lateral knee stability has an effect on progression to knee arthroplasty [10,11,12]. Since TFJD leads to greater lateral gapping, a dynamic varus knee may ensue, thus leading to worse mid- and long-term outcomes and lower CWHTO survival rates. This is especially so if no overcorrection is achieved after surgery.

The aim of this study was to determine whether two different ways of surgically manipulating the TFJ affect HTO survival rates and if this has an influence on the clinical medium and long-term results in those patients who have undergone a CWHTO. The main hypothesis of this study is that a TFJD increases lateral compartment gapping as it leads to joint instability, worsening CWHTO outcomes and could possibly reduce the survivorship of this procedure.

## 2. Methods

It is a retrospective cohort study that included all lateral CWHTO performed by seven expert knee surgeons in the same center from January 2005 to December 2018. For the study, patients were divided into two cohorts depending on the technique used over the proximal TFJ: group 1 in the case of FHO and group 2 in the case of TFJD. The study was approved by the ethics committee of our institution (2019/8762/I).

### 2.1. Subjects

The indication for osteotomy was medial compartment pain in a relatively young and active patient with medial varus knee osteoarthritis with or without prior medial meniscectomy, excluding patients with post-traumatic osteoarthritis or other previous surgeries that were not a simple meniscectomy.

Preoperative demographic variables like age, gender, laterality and BMI were collected. 

### 2.2. Surgical Procedure

All patients underwent a lateral CWHTO with the same surgical approach, an osteotomy at the same level under fluoroscopy control as well as fixation. It was done with the Natural-Knee^®^ High Tibial Osteotomy (HTO) System (Zimmer^®^, Warsaw, IN, USA) or conventional Coventry staples. It depended on the surgeon’s preference. In all cases, a prior arthroscopy was performed with or without medial meniscectomy and knee compartment revision prior to osteotomy. The objective was the correction of the mechanical axis at the Fujisawa point. Depending on surgeon preference, FHO or TFJD was performed (Figure 1) prior to tibial osteotomy. 

### 2.3. Radiological Evaluation

A weight-bearing full-leg length x-ray was collected preoperatively, one year after surgery, and at the end of the follow-up for all the patients to measure the femorotibial angle (FTA) using the mechanical axis (hip-knee-ankle angle). The Rosenberg view was used to evaluate medial compartment osteoarthritis using the Ahlbäck scale. The radiological study was performed on the PACS computer system (Picture Archiving and Communication System).

### 2.4. Clinical Evaluation

The Knee Society Score (KSS) [13], specifically the Spanish version [14], was used for clinical and functional evaluation preoperatively and at the end of the follow-up (range: 24 to 180 months). The five-point Likert scale was used to measure satisfaction [15]. Moreover, patients were asked if they would undergo surgery again knowing the result obtained (yes/no) at the end of follow-up.

Surgical complications related to the procedure were collected. TKR was considered the endpoint to analyze the survival rate of CWHTO. 

### 2.5. Statistical Analysis

For the initial statistical analysis, quantitative variables were described with mean and standard deviation. Qualitative variables were described with frequency tables (number and percentage). Between-group comparisons were tested with the Mann−Whitney U test. Kaplan–Meier survival curves were performed for TKR events relative to the proximal FTJ. The log-rank test was performed to check for differences between survival curves. Additionally, between-curve differences were checked at several time points throughout the follow-up. STATA version 15.1 (StataCorp, College Station, TX, USA) was used for statistical analysis. *p*-values ≤ 0.05 were considered statistically significant.

A sub-analysis of both cohorts was performed on patients who had not been FTA overcorrected after surgery (postoperative FTA ≤ 180°, continuous loading in varus). 

## 3. Results

A total of 290 knees of 272 patients were operated on, 6 died, and 48 were lost to follow-up. Finally, 230 knees of 216 patients were analyzed and follow-up ranged from 24–180 months. 

There were 105 FHO knees and 125 proximal TFJD knees. Table 1 gives a demographic data summary and the preoperative radiological evaluation of both groups. No preop differences were found.

No differences were found in terms of complications, reoperation for infection or acute osteotomy failure, considered loss of correction in the first 3 months (Table 2).

Table 3 shows comparative clinical and radiological results. No differences were seen in the KSS. Additionally, there was no statistically significant difference in the conversion to TKR at the end of follow-up. In general, the patients were satisfied with the outcome of the procedure and the majority of them would be operated on again independently of the technique used for the TFJ. A statistically significant difference was found in the mean follow-up time, which was longer in the case of FHO. There was also a statistically significant difference on the Ahlbäck scale at the end of the follow-up with a greater progression in the case of TFJ dislocation.

A non-significant trend of less survival in the TFJD group was found (Figure 2) even though there was a tendency towards a significant difference at a longer follow-up time (Table 4).

The first postoperative control radiography with an FTA angle ≤ 180° (non-overcorrected) was done on 52 knees from the FHO group (49.5%) and 58 of the TFJD group (46.4%). In the analysis of the subgroups of non-overcorrected patients, in addition to continuing to observe the difference in the Ahlbäck scale at the end of follow-up, a significant difference was found at the TKR endpoint at the end of follow-up. A TKR was required for 12.8% of the FHO patients with a mean of 88.8 months from CWHTO compared to 26.8% with a mean of 54.9 months in the case of TFJD (Table 5). 

In patients who had not required TKR, no differences were found in terms of clinical outcomes, changes in the FTA during follow-up or relative to satisfaction.

In the subgroup analysis of non-overcorrected patients, significant differences were found in osteotomy survival depending on the technique used on the proximal tibiofibular joint (Figure 3). If we analyze survival by time ranges, we find a significant difference after 60 months of follow-up (Table 6). 

## 4. Discussion

The main finding of this study is that, in the medium and long-term, patients who have undergone CWHTO in association with a TFJD progress to TKR conversion in less time when compared to patients who undergo it with an associated FHO. This finding is quite significant in those patients who are undercorrected after HTO and that present FTA is still in varus after surgery. We also found a significant progression on the Ahlbäck scale, being also worse in its progression in those patients with TFJD.

Catherine Hui et al. analyzed the survival rate of CWHTO in almost 400 patients and obtained results similar to those arrived at in our study. They observed 95% at the 5- and 79% at 10-year follow-up. In our series, we obtained 93% and 87% at 5 and 10 years in association with FHO and 89% and 73.6% in cases of TFJD.

Contrary to the initial hypothesis, we did not find relevant clinical differences in patients who completed follow-up without TKR. They presented the same values in the KSS R and KSS F independently of whether a TFJD or FHO had been done. In the medium and long-term, the progression of the FTA was similarly independent of the procedure in proximal TFJ. Therefore, we could not associate a lower survival for HTO to a progression of varus deformity.

We did not find prospective or retrospective clinical studies that evaluated different ways of treating the proximal tibiofibular joint in CWHTO to compare our results to.

Torres et al. [9] carried out a prospective randomized study performing TFJD or FHO in CWHTO, assessing lateral stability by stress radiology and functional results with KSS R and KSS F at 1 year follow-up. They concluded that lateral laxity increases with PDJT although without clinical differences at one year in both procedures, suggesting that it is possible clinical differences could appear in the long term due to the residual instability observed, especially in patients with postoperative varus. Continuous stress in the lateral compartment with insufficient stability could result in a greater progression of the residual varus deformity. We did not find these clinical differences, and we cannot affirm that the lower survival for HTO in the TFJD group is due to this fact, since we did not study the stability of the lateral ligament complex in our analysis, and we did not observe a different progression of the varus in any case.

A theory to explain the lower survival rate of the TFJD group is that lateral laxity may cause lower survival for HTO in some patients, an increase in arthropathy in load (change in the Ahlbäck scale) and a greater conversion to TKR in a dynamic way without objective progression of the deformity of the FTA in radiological study. Another theory is that these differences are due to the beneficial therapeutic effect of the proximal fibula osteotomy independently added to the HTO. In the last 5 years, some studies have been published, the majority biomechanical [16,17,18,19,20] and clinical [21,22,23,24,25,26,27] trials mainly from the Asian continent, which indicate that isolated FHO can reduce the pressure of the medial femorotibial compartment, improving the symptoms of varus knee osteoarthritis.

In 2015, Yang et al. [16] published the first clinical series with 156 cases of isolated proximal fibular osteotomy with 2 years of follow-up, and they conclude that this procedure can significantly improve both the radiographic appearance and function of the affected knee joint and can also achieve long-term pain relief. They affirm that this procedure may be an alternative treatment option for medial compartment osteoarthritis like HTO or TKR. Baldini et al. [26] presented a study in 2018 in which ten matched pairs of cadaver legs were tested under compression to 1.1 times the body weight comparing an intact knee with proximal fibular osteotomy at 0°, 15° and 30° of flexion and concluded that the proximal fibular osteotomy decreases the pressure in the medial compartment of the knee, which may reduce knee pain and improve function in patients with medial compartment knee osteoarthritis. In a 2019 retrospective study with radiographic analysis of 560 knees, Wang et al. [27] concluded that FHO produces a reduction in knee adduction moments and rebalances the biceps-proximal fibula-peroneus longus complex reducing pressure on the medial compartment. These biomechanical, radiological and clinical studies support the theory that the direct effect of FHO may be the cause of the longer survival of this group in our series, but there are no prospective studies that provide greater clinical evidence for this claim.

This study has several limitations: First, it was a retrospective non-randomized one. However, both groups were similar in terms of demographic, clinical and radiographic variables. Second, the average follow-up is only 109.8 months for FHO and 87.3 months in TFJD group. Therefore, we could not analyze the different effects of both techniques beyond 15 years of follow-up, but the survival analysis seems to show differences in the first 5 years and that these differences are maintained over time.

In conclusion, TFJD associated with CWHTO shows an increase in the conversion to TKR at medium and long-term follow-up with lower HTO survival than CWHTO associated to FHO, especially in patients with a postoperative FTA ≤ 180° (non-overcorrected). Nevertheless, in the patients not converted to TKR clinical, radiological and satisfaction results are similar for both groups.

## Figures and Tables

**Figure 1 jcm-10-02743-f001:**
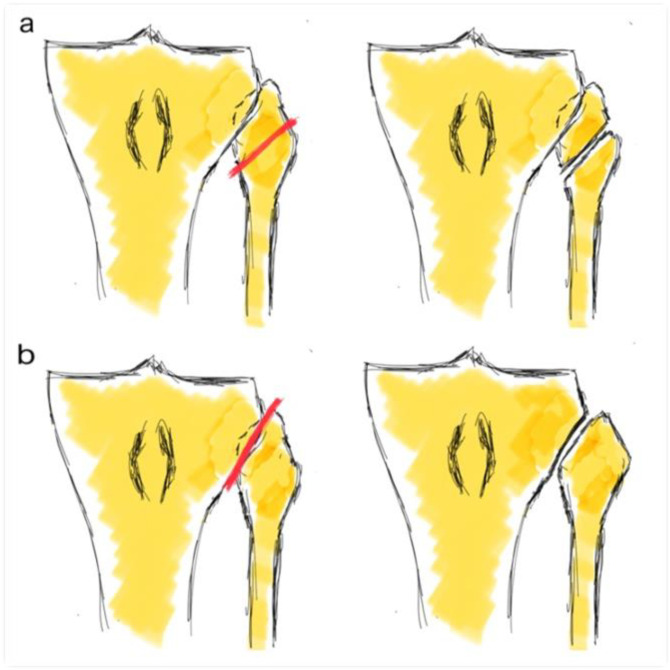
Fibular head osteotomy (**a**) and tibiofibular joint dislocation (**b**). Figure reproduced from [9].

**Figure 2 jcm-10-02743-f002:**
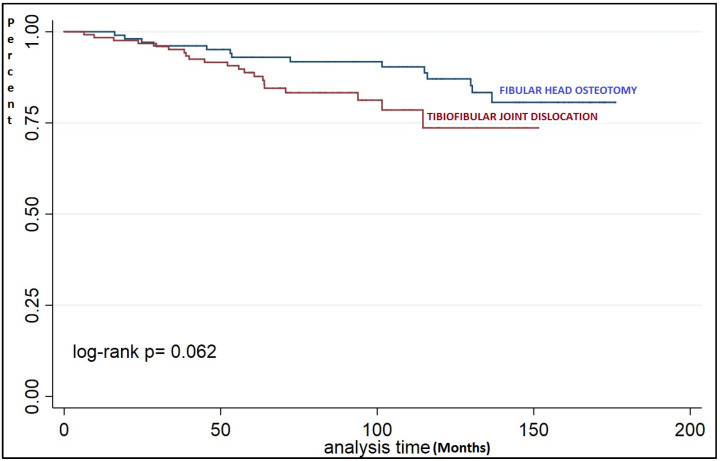
Survival analysis groups (log-rank test).

**Figure 3 jcm-10-02743-f003:**
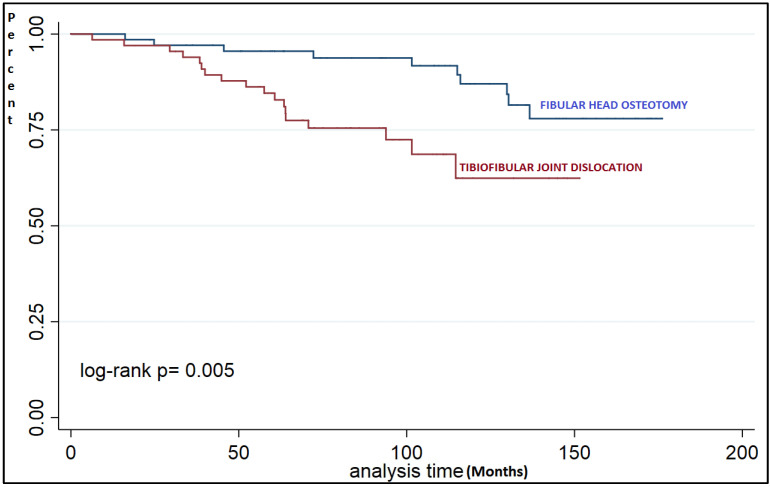
Survival analysis of non-overcorrected patients, FTA ≤ 180° (log-rank test).

**Table 1 jcm-10-02743-t001:** Demographic data, preoperative KSS knee and function, femorotibial angle (FTA) and Ahlbäck scale.

Title	Group 1. FHO (*n* 105)	Group 2. TFJD (*n* 125)	*p*-Value
Age (years)	53.2 (SD 9.2)	54.9 (SD 8.4)	n.s.
Biological sex (male/female)	70/35	81/44	n.s.
Body mass index (Kg/m^2^)	30.2 (SD 7.4)	29.2 (SD 4.6)	n.s.
Side (right/left)	47/58	64/61	n.s.
Previous FTA	170.9° (SD 3.6)	171.6° (SD 2.3)	n.s.
Ahlbäck preop	1.9 (SD 0.3)	2.1 (SD 0.4)	n.s.
KSS knee	53.9	54.7	n.s.
KSS function	69.8	71.3	n.s.

**Table 2 jcm-10-02743-t002:** Infection and acute osteotomy fail.

Title	Group 1. FHO (*n* 105)	Group 2. TFJD (*n* 125)	*p*-Value
Infection	7	9	n.s.
Osteotomy failure	3	4	n.s.

**Table 3 jcm-10-02743-t003:** Results summary in function of the FHO or TFJD group: clinical results (KSS R and KSS R), radiological 1 year follow-up (FTA 1y FU and Ahlbäck 1y FU), radiological final follow-up (FTA final FU and Ahlbäck final FU), mean of follow-up, FTA undercorrected, number of TKR (TKR final FU), time until TKR since HTO in months (Time to TKR), Likert scale of satisfaction (1–5) and if the patient would be operated on again (Would you repeat surgery?, yes/no).

Title	Group 1. FHO (*n* 105)	Group 2. TFJD (*n* 125)	*p*-Value
Mean Follow-Up (FU)	109.8 months (SD 45.6)	87.3 months (SD 32.3)	0.01
KSS R	86.5 (SD 15.9)	84.3 (SD 15.8)	0.33
KSS F	79.7 (SD 23.2)	77.6 (SD 20.6)	0.31
FTA 1y FU	178.9° (SD 4.8)	179.9° (SD 3.8)	0.12
FTA final FU	178.7° (SD 4.9)	179.5° (SD 4.2)	0.11
Undercorrected FTA ≤ 180°	52 (49.5%)	58 (46.4%)	0.13
Ahlbäck 1y FU	2.07 (SD 0.6)	2.27 (SD 0.4)	0.06
Ahlbäck final FU	2.21 (SD 0.5)	2.55 (0.5)	0.02 *
TKR final FU	14 (13.3%)	21 (16.8%)	0.58
Time to TKR	74.4 months (SD 45.5)	61.36 months (SD 28.6)	0.16
Satisfaction (Likert 1–5)	3.8 (SD 1.1)	3.8 (SD 1.2)	0.97
Would you repeat surgery? (yes/no)	90/15 (85.7%)	102/23 (81.6%)	0.55

* statistically significant difference.

**Table 4 jcm-10-02743-t004:** Differences in survival percentages at different time points.

Time Follow-Up (Months)	Group 1. FHO (*n* 105)	Group 2. TFJD (*n* 125)	*p*-Value
24	98%	96.8%	0.53
36	96.1%	95.1%	0.71
48	95.1%	91.6%	0.29
60	93%	88.8%	0.28
120	87%	73.6%	0.07

**Table 5 jcm-10-02743-t005:** Results summary relative to the FHO or TFJD group in non-overcorrected patients, FTA ≤ 180°: clinical results (KSS R and KSS R), radiological 1 year follow-up (FTA 1y FU and Ahlbäck 1y FU), radiological final follow-up (FTA final FU and Ahlbäck final FU), number of TKR (TKR final FU), time until TKR since HTO in months (Time to TKR), Likert scale of satisfaction (1–5) and if the patient would be operated again (Would you repeat surgery?, yes/no).

Title	Group 1. FHO (*n* 52)	Group 2. TFJD (*n* 58)	*p*-Value
KSS R	86.9 (SD 15.7)	83.3 (SD 17.4)	0.4
KSS F	80.7 (SD 24.8)	78.1 (SD 20.9)	0.33
FTA 1y FU	176.1° (SD 3.7)	177.1° (SD 2.9)	0.24
FTA final FU	175.6° (SD 3.9)	176.5° (SD 4.2)	0.48
Ahlbäck 1y FU	2.11 (SD 0.6)	2.25 (SD 0.5)	0.32
Ahlbäck final FU	2.27 (SD 0.5)	2.6 (0.5)	0.04 *
TKR final FU	9 (17.3%)	18 (31%)	0.04 *
Time to TKR (months)	88.8 (SD 45.7)	54.9 (SD 28.3)	0.03 *
Satisfaction (Likert 1–5)	3.9 (SD 1.1)	3.7 (SD 1.3)	0.67
Would you repeat surgery? (yes/no)	58/12 (82.8%)	55/12 (82%)	0.59

*:. statistically significant difference

**Table 6 jcm-10-02743-t006:** Differences in the percentage of survival at different time points in non-overcorrected patients.

Time Follow-Up (Months)	Group 1. FHO (*n* 52)	Group 2. TFJD (*n* 58)	*p*-Value
24	98.5	97	0.54
36	97	93.9	0.38
48	95.5	87.8	0.14
60	95.5	84.6	0.03
120	82.6	62.4	0.01

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
