# Peer review of "Proximal Tibiofibular Dislocation in Closing-Wedge High Tibial Osteotomy Increases the Risk of Medium and Long-Term Total Knee Replacement"

_jcm, 2021, doi:10.3390/jcm10132743_

Round 1

Reviewer 1 Report

The present study, based on clinical retrospective data, showcases thoroughly that tibiofemoral joint dislocation as a method to alleviate medial knee loading hence local deterioration of cartilage homeostasis, is related to an increase in total knee arthroplasty interventions with lower osteotomy survival. Hence, the authors suggest that a fibular head osteotomy may be a better surgical choice to decrease risk of surgical knee replacement. I believe that the paper is very well presented and discussed, hence I do not have any special comments to make.

Author Response

Thank you very much for your review.

Reviewer 2 Report

With the manuscript entitled "Proximal tibiofibular dislocation in the closing-wedge high tibial osteotomy increases the risk of medium and long-term total knee replacement" the authors present an analysis of two different closing-wedge high tibial osteotomy (HTO) methods: One combined with fibular head osteotomy (FHO) and the other with tibio-fibular dislocation (TFD). 

Although I am not a domain expert (surgeon) the article was of interest to me. However, I cannot judge on the completeness of the related work. The motivation as well as the study design seems to be filtered by the outcome. I would prefer seing a complete data analysis with conclusions drawn thereof instead of a reduced analysis with respect to a subgroup.

Reading the introduction the first time I was a bit confused. I would advise the authors using a consistent notation for TFD throughout the entire paper. At the moment TFJD, FTJD, and FTD is mixed. In addition, since open-wedge procedures are not taken into account for this analysis, I would not thematize this procedure at all in order to keep the story focused and concise. Try to avoid unnecessary repetitions.

On the basis of clinical full-leg AP radiographs 225 knees were evaluated, 105 FHO cases (group 1) and 125 TFD cases (group 2). Please try to use the group assignment in a consistent manner. Group 1 and group 2 were interchanged inbetween (see first paragraph of Methods).

According to the authors, "the aim of this study was to determine whether two different ways of surgically manipulating the TFJ affect HTO survival rates and whether this has an influence on clinical medium and long-term results in those patients who have undergone a CWHTO".

The analysis itself is an evaluation of various indicators (measures) after the respective HTO treatment over a period of 10 years with its endpoint defined as a total knee replacement (TKR). The analysis of the progression of indicators over the complete follow-up would be of interest for all patients independent whether or not they reach the endpoint or having been under- or overcorrected.

The authors focused on undercorrected cases wrt. the femoro-tibial angle (FTA). This reduces the amount of knees by approx. 50%. Although I do understand the reasoning behind this, I would still prefer seeing the analysis for all 3 subgroups (independent of the TKR endpoint) within the methods, i.e., (1) all knees, (2) non-overcorrected, and (3) overcorrected ones. The development of your measures for overcorrected cases as well as for patients that did not reach the TKR endpoint would give some additional insight to the readers so please provide additional tables/plots for all measures. You may discuss the additional tables concluding what you found.

In Figures 2 and 4, the differences between the two lines are hard to see. Either use a solid and a dashed line or write the respective meaning (FHO, TFD) close to the line. Also, please add units to the axes (months, percent),
Could you also mark in Figure 4 the regions where the deviation is significant? Tables 4 and 6 show just 5 time points - the graph, however, seems to contain more time-points. Why aren't these time points listed in the tables? If possible, please provide the complete data in a table within an appendix or as supplementary material.

Within your discussion you mention that you did not find relevant clinical differences in patients who completed follow-up without TKR. However, I would expect seing some trends in the data that you should also present and discuss. The analysis should have been performed on the full dataset and the complete results should be presented, giving a reader the opportunity to get the full picture. If the tables and plots then show that there is no or just a small trend in changes for patients who did not reach TKR this may lead to or confirm your hypothesis. 

Some remarks on details below:

Notation: Page 1, line 22 (1:22)
 (1:22): "... between ... AND ..."
 (1:22): "... were placed ..."
 (3:105): "prior to" - I assume "after" tibial osteotomy
 (4:142): missing space between "Table 1" and "gives"
 (7:236): check for font color "concluded that the"

Table 1: instead of "gender" "biological sex" might be more appropriate

Table 2: are the numbers 105 and 125 reduced by these cases?

 (6:197): please cite Hui et al. correctly [11]. 

Author Response

Thank you very much for your review. Below I try to respond to your comments and I attach the corrected document after your evaluation

Point 1.

The motivation as well as the study design seems to be filtered by the outcome. I would prefer seing a complete data analysis with conclusions drawn thereof instead of a reduced analysis with respect to a subgroup.

Response point 1.

It is true that the retrospective study is designed based on a previous prospective study in which we observed objective lateral laxity after a proximal tibiofibular dislocation with no clinical differences one year after follow-up (Reference 9). From there, we asked ourselves if there would be differences in the longer term, we analyzed 2 cohorts, functional results and survival (end point knee arthroplasty). We analyze, on the one hand, all the patients (with complete data) and on the other, the non-overcorrected, for two reasons, because on the one hand it is already known that they are the ones that can have the worst result (in fact, 27 out of 35 total knee prostheses were given in undercorrected patients in our series) and, on the other, it is in those that lateral stability becomes more important when continuing to function in varus and may influence the technique used on the proximal tibiofibular joint.

Point 2.

Reading the introduction the first time I was a bit confused. I would advise the authors using a consistent notation for TFD throughout the entire paper. At the moment TFJD, FTJD, and FTD is mixed. In addition, since open-wedge procedures are not taken into account for this analysis, I would not thematize this procedure at all in order to keep the story focused and concise. Try to avoid unnecessary repetitions.

Response point 2.

Revised, TFJD in all cases. In the introduction we named the open wedge osteotomy because it is the most commonly used osteotomy at the moment, but we want to emphasize that in recent studies the results are the same as those of the closed wedge and for this reason it seems to us that our study may be relevant.

Point 3.

On the basis of clinical full-leg AP radiographs 225 knees were evaluated, 105 FHO cases (group 1) and 125 TFD cases (group 2). Please try to use the group assignment in a consistent manner. Group 1 and group 2 were interchanged inbetween (see first paragraph of Methods).

Response point 3.

Sorry, but I have not found 225 knees in the document, there are 230 knees, 105 in group 1 and 125 in group 2. I don't know if you have reviewed an outdated document. First paragraph of methods revised, it was badly written and changed as you point out

Point 4.

The analysis itself is an evaluation of various indicators (measures) after the respective HTO treatment over a period of 10 years with its endpoint defined as a total knee replacement (TKR). The analysis of the progression of indicators over the complete follow-up would be of interest for all patients independent whether or not they reach the endpoint or having been under- or overcorrected.

Response point 4.

Regarding analysis patients overcorrected, we answer in point 1. On the other hand, the analysis of patients with TKR after osteotomy we believe is the objective of another study.

Point 5.

The authors focused on undercorrected cases wrt. the femoro-tibial angle (FTA). This reduces the amount of knees by approx. 50%. Although I do understand the reasoning behind this, I would still prefer seeing the analysis for all 3 subgroups (independent of the TKR endpoint) within the methods, i.e., (1) all knees, (2) non-overcorrected, and (3) overcorrected ones. The development of your measures for overcorrected cases as well as for patients that did not reach the TKR endpoint would give some additional insight to the readers so please provide additional tables/plots for all measures. You may discuss the additional tables concluding what you found.

Response point 5.

We analyze 1. All Knees and 2. Non-overcorrected for the reason explained in point 1

Point 6.

In Figures 2 and 4, the differences between the two lines are hard to see. Either use a solid and a dashed line or write the respective meaning (FHO, TFD) close to the line. Also, please add units to the axes (months, percent),
Could you also mark in Figure 4 the regions where the deviation is significant? Tables 4 and 6 show just 5 time points - the graph, however, seems to contain more time-points. Why aren't these time points listed in the tables? If possible, please provide the complete data in a table within an appendix or as supplementary material.

Response point 6.

The figures have been modified according to the reviewer's notes. Regarding the cut-off time points, these have been chosen as representative at 2, 3, 4, 5 and 10 years of follow-up.

Point 7.

Within your discussion you mention that you did not find relevant clinical differences in patients who completed follow-up without TKR. However, I would expect seing some trends in the data that you should also present and discuss. The analysis should have been performed on the full dataset and the complete results should be presented, giving a reader the opportunity to get the full picture. If the tables and plots then show that there is no or just a small trend in changes for patients who did not reach TKR this may lead to or confirm your hypothesis.

Response point 7.

I do not understand your appreciation, the analysis has been carried out with all the data without finding relevant clinical differences as indicated.

SOME REMARKS

Notation: Page 1, line 22 (1:22)
 (1:22): "... between ... AND ..." Reviewed
 (1:22): "... were placed ..."
 (3:105): "prior to" - I assume "after" tibial osteotomy No, the arthroscopy is prior to tibial osteotomy to check the correct indication and for meniscectomy in some cases
 (4:142): missing space between "Table 1" and "gives" Reviewed
 (7:236): check for font color "concluded that the" Reviewed, it was the english edition

Table 1: instead of "gender" "biological sex" might be more appropriate Changed

Table 2: are the numbers 105 and 125 reduced by these cases? No

 (6:197): please cite Hui et al. correctly [11] Corrected
